# How many days of continuous physical activity monitoring reliably represent time in different intensities in cancer survivors

Benedikte Western[1]*, Ingrid Demmelmaier[1,2], Ingvild Vistad[3,4], Bjørge Herman Hansen[1], Andreas Stenling[1,5], Hege Berg Henriksen[6], Karin Nordin[2], Rune Blomhoff[6,7], Sveinung Berntsen[1,2]

1 Department of Sport Science and Physical Education, University of Agder, Kristiansand, Norway, 2 Department of Public Health and Caring Sciences, Uppsala University, Uppsala, Sweden, 3 Department of Clinical Science, University of Bergen, Bergen, Norway, 4 Department of Obstetrics and Gynecology, Sørlandet Hospital, Kristiansand, Norway, 5 Department of Psychology, Umeå University, Umeå, Sweden, 6 Department of Nutrition, Institute of Basic Medical Sciences, University of Oslo, Oslo, Norway, 7 Division of Cancer Medicine, Oslo University Hospital, Oslo, Norway

* benedikte.western@uia.no

**Data Availability Statement:** All relevant data are within the paper and its Supporting Information files.

## Abstract

### Background

Physical activity (PA) monitoring is applied in a growing number of studies within cancer research. However, no consensus exists on how many days PA should be monitored to obtain reliable estimates in the cancer population. The objective of the present study was to determine the minimum number of monitoring days required for reliable estimates of different PA intensities in cancer survivors when using a six-days protocol. Furthermore, reliability of monitoring days was assessed stratified on sex, age, cancer type, weight status, and educational level.

### Methods

Data was obtained from two studies where PA was monitored for seven days using the SenseWear Armband Mini in a total of 984 cancer survivors diagnosed with breast, colorectal or prostate cancer. Participants with ≥22 hours monitor wear-time for six days were included in the reliability analysis (n = 736). The intra-class correlation coefficient (ICC) and the Spearman Brown prophecy formula were used to assess the reliability of different number of monitoring days.

### Results

For time in light PA, two monitoring days resulted in reliable estimates (ICC >0.80). Participants with BMI ≥25, low-medium education, colorectal cancer, or age ≥60 years required one additional monitoring day. For moderate and moderate-to-vigorous PA, three monitoring days yielded reliable estimates. Participants with BMI ≥25 or breast cancer required one additional monitoring day. Vigorous PA showed the largest within subject variations and reliable estimates were not obtained for the sample as a whole. However, reliable estimates

**Funding:** The author(s) received no specific funding for this work.

**Competing interests:** The authors have declared that no competing interests exist.

were obtained for breast cancer survivors (4 days), females, BMI $\geq$30, and age <60 years (6 days).

## Conclusion

Shorter monitoring periods may provide reliable estimates of PA levels in cancer survivors when monitored continuously with a wearable device. This could potentially lower the participant burden and allow for less exclusion of participants not adhering to longer protocols.

## Introduction

Physical activity (PA) may improve health outcomes in cancer survivors, including fatigue, anxiety, depressive symptoms, physical functioning, and health-related quality of life [1]. As the field of exercise oncology is expanding, PA levels before, during, and after cancer treatment are increasingly measured and reported in cancer research [2, 3].

A wide range of instruments are currently used for measuring PA. Questionnaires are the most common approach for collect PA data as they are cost-effective and can be distributed to large samples [4]. However, self-reported PA is at risk of recall-, misclassification-, and social desirability bias, and cancer survivors are likely to overreport their activity level when using questionnaires [5, 6]. Objective assessments, in the form of wearable PA monitors, can provide more reliable PA estimates compared to questionnaires, but is also not without limitations. How PA data is collected and processed can impact the quality of the acquired data, and methods have been found inconsistent across studies of cancer survivors, especially regarding the number of days to monitor [7]. Furthermore, required monitor wear-time, encompassing both the number of days and hours per day to measure, are merely defined in half of the studies with the purpose of quantifying PA in cancer survivors through accelerometers [7]. While standardization of monitoring protocols can be advantageous for comparison and replication, it has been argued that the appropriate number of days to monitor PA is dependent on the research question [8, 9]. Generally, large sample sizes have been shown to require fewer monitoring days and produce lower standard errors of the mean (SEM), thus providing more reliable estimates, compared to smaller sample sizes with numerous monitoring days [9].

Extensive monitoring periods may be burdensome for some participants and could potentially lead to non-consent of study participation and non-adherence to monitoring protocols [10, 11]. The burden of study participation may be greater in persons with medical conditions also affected by the disease burden compared to healthy adults. Ideally, the monitoring protocol with the least participant burden and most reliable estimates would be the most appropriate. The number of days PA should be monitored to reliably represent time in PA intensities can be found by assessing the intra-individual and inter-individual variability in PA across monitoring days (i.e., the within- and between-subject variation). With increased day-to-day variation in PA within subjects, more monitoring days would be needed for reliable estimates representing a certain point in time.

In the general population, the reliability of number of monitoring days have been assessed in numerous studies based on various timeframes, daily wear-time, and sample sizes [12–23]. However, the results are inconclusive, reporting reliable estimates of moderate-to-vigorous PA (MVPA) with 2–10 monitoring days. The ambiguous results may relate to how many days' variability is considered, thus, the respective timeframes serving as the foundation for the reliability estimates. Also, the varying daily monitor wear-times ranging from 8–24 hours can

impact the variability in measured PA. Continuous wear of the monitors have only been assessed in a few studies, with the purpose of obtaining estimates representing absolute time in PA, limiting variance caused by differing wear-times [16, 17, 23]. With increasing technological developments of wearable PA devices including longer battery life and more comfortable designs and ease of use, allowing for continuous wear and monitoring, there is a need for studies utilizing such wear-time protocols. Reliability assessments of PA estimates in cancer survivors are scarce and no consensus exists on how many days to monitor, which have led to considerable inconsistency in monitoring protocols [2, 3, 7]. Three monitoring days have been found reliable in representing time in MVPA in colorectal cancer survivors >6 months post-surgery, with an accelerometer worn during waking hours [24]. However, no study has made these assessments for different PA intensities in a mixed sample of cancer survivors using continuous monitor wear-time, nor assessed whether participant characteristics impact the reliability.

The aim of the present study was to determine the minimum number of monitoring days for reliable estimates of time in different PA intensities in cancer survivors, using a continuous wear-time protocol. Furthermore, the reliability was assessed stratified on sex, age, diagnosis, weight status, and educational level.

## Material and methods

### Participants and study design

In the present study we harmonized baseline data from the Phys-Can study [25] and the CRC-NORDIET study [26]. The current hypotheses and statistical analyses were not prospectively registered, rather, application for use of the data was sent to the respective studies and processed and approved by the boards.

The harmonized dataset consisted of 984 participants diagnosed with either breast, colorectal or prostate cancer, stages I-III. The CRC-NORDIET study included participants with colorectal cancer who completed baseline 2–9 months post curative surgery (median 5.3 months), with approximately 1/5 also receiving post-surgery chemotherapy. The Phys-Can study included participants with colorectal, breast or prostate cancer who completed baseline before starting neoadjuvant or adjuvant therapy. In both studies, PA levels were measured using the SenseWear™ Armband Mini (SWAM) (BodyMedia Inc. Pittsburgh, PA, USA) and the same monitoring protocol was followed (i.e., the same instructions on how to wear the monitor and the continuous wear throughout seven days were provided).

### Physical activity instrument

The SWAM is a multi-sensor device containing a tri-axial accelerometer and sensors measuring heat flux, galvanic skin response, skin temperature, and near-body ambient temperature. The SWAM has been validated for estimating total energy expenditure and showed promise in accurately measuring daily energy expenditure under free-living conditions as well as resistance training [27–29]. The original Sensewear Armband has previously been tested in cancer populations [30]. The SWAM was placed on the non-dominant upper arm.

### Data management

A valid day of PA monitoring was defined as ≥22 hours wear of the monitor. Currently, there are no consensus on how long a monitor should be worn each day to produce reliable estimates. Thus, 22 hours representing >90% of a day, was chosen as we wanted to use continuous monitoring and absolute time in PA intensities, allowing for short periods of removal.

Raw data was handled using software developed by the manufacturer (Sensewear Professional Research Software Version 8.1, BodyMedia Inc., Pittsburgh, PA, USA). Metabolic equivalents (METs) were calculated based on the accelerometer and temperature sensors through algorithms in the SWAM software. METs were used for representing time in PA intensities. Light intensity PA (LPA) was defined as METs between 1.5 and 3. MET values of 3–6 corresponded with moderate intensity PA (MPA), while vigorous intensity PA (VPA) was established for MET values >6. Thus, MVPA corresponded with METs ≥3.

Within the monitoring week, the first day showed inadequate wear-time across the sample, as it was usually the day SWAM was administered to the participants. Thus, the first monitoring day was excluded from analyses and six days served as the criterion. For participants with more than six valid days, the first consecutive six days with sufficient wear-time were used.

The sample was further stratified on sex (male, female), age (<60 and ≥60), diagnosis (colorectal, breast, and prostate cancer), weight status (body mass index (BMI) <25, ≥25 <30, and ≥30), and educational level (low-medium and high). Low-medium education included primary and secondary school, while higher education included education at college or university level. Details on how these variables were measured have been reported elsewhere [25, 26]. Participant characteristics used for stratification were chosen based on their availability within the dataset, as well as their theoretical relevance related to PA level, and were hypothesized to also have potential impact on the variance in PA. No category representing underweight BMI was made as only eight subjects were below BMI 18.5 and were thus included in BMI <25.

## Ethic statement

Written informed consent was obtained from all participants enrolled in the two studies. The Phys-Can study was approved by the Regional Ethical review Board in Uppsala, Sweden (protocol approval 2014/249) and registered in ClinicalTrials.gov (NCT02473003, Oct 2014). The CRC-NORDIET study was approved by the Reginal Committees for Medical and Health Research Ethics, Norway (protocol approval 2011/836), and the data protection officials at Oslo University Hospital and Akershus University Hospital, and registered in ClinicalTrial.gov (NCT01570010).

## Statistical analyses

Differences in characteristics between participants with and without six valid monitoring days were assessed with independent sample t-tests for continuous variables, and the Pearson Chi-Square test for categorical variables.

The intra-class correlation coefficient (ICC) was used to study the variance in PA across the six days. The coefficient for "single measures" gives the relative contribution of inter-individual variance on the total variance and indicate the reliability of using one monitoring day to represent the monitoring period. The Spearman Brown prophecy formula for interrater reliability was applied to calculate the reliability of using the average of an increasing number of days to represent PA levels based on the measured six days [31–33]. The Spearman Brown prophecy formula was expressed as $((k \times r) \div (1+(k\text{-}1) \times r))$ where $k$ is the number of days and $r$ is the single measures coefficient [31, 32]. An ICC >0.80 was considered sufficient for reliable estimates [34].

Results from the one-way random (ICC(1)), the two-way random absolute agreement (ICC(A,1)) and two-way random consistency (ICC(C,1)) were compared to assess bias contribution to the total variance (Table 1 in S1 File) [35]. Bias contribution, as well as intra-individual and inter-individual contribution to the variance were calculated based on mean squares from the

ICC(A,1), and presented in (Table 3 in S1 File) [35]. Bias was found negligible (<1% of the total variance) and coefficients consistent across models.

Results were considered statistically significant for p-values <0.05. Analyses were conducted using SPSS 25 (IBM Corp. Released 2017. IBM SPSS Statistics for Windows, Version 25.0. Armonk, NY: IBM Corp.), while the Spearman Brown formula was calculated by hand.

## Results

Of the 984 cancer survivors, 736 participants (74.8%) had ≥22 hours daily SWAM wear-time for six days or more and were included in the reliability analyses (Table 1). Their mean age and standard deviation (±SD) were 62.6 years (±10.5), with a mean BMI of 26.4 (±4.6). For the 248 excluded cancer survivors, age was significantly lower (59.8 years ±11.6, p<0.01) and BMI similar (25.8 ±4.3, p = 0.058). Various descriptive data were missing from 46 participants across the two groups for unknown reasons. Excluding them from the analyses did not alter the results and they were kept in the present analyses.

The ICC absolute agreement [95% confidence interval] for single measures was 0.690 [0.660, 0.716] for LPA, 0.606 [0.576, 0.636] for MPA, 0.378 [0.345, 0.412] for VPA, and 0.610 [0.580, 0.639] for MVPA.

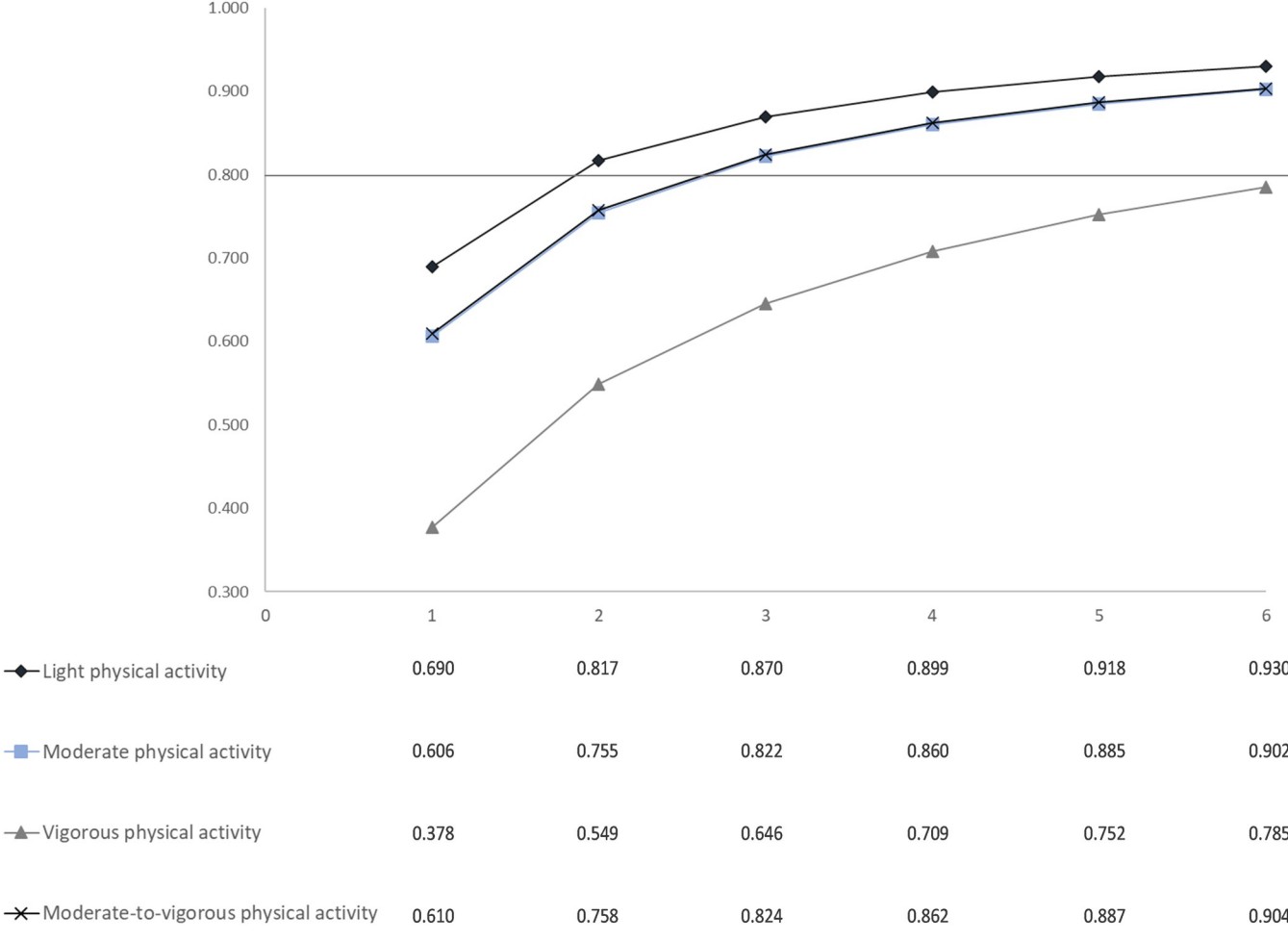

**Fig 1. Number of monitoring days with the corresponding intra-class correlation coefficient (ICC) based on the Spearman Brown formula for each physical activity intensity (n = 736).**

**Table 1.  Descriptive statistics of cancer survivors with six valid measuring days (included) and without six valid days (excluded).**

| Characteristics | Included | Excluded | Difference |
|---|---|---|---|
| | n = 736 | n = 248 | |
| | n (%) | n (%) | |
| **Sex** | | | p = 0.827 |
| Male | 259 (35.2) | 80 (32.3) | |
| Female | 464 (63.0) | 163 (65.7) | |
| **Age** | | | p = 0.011* |
| <60 years | 250 (34.0) | 106 (42.7) | |
| ≥60 years | 473 (64.3) | 137 (55.2) | |
| **Diagnosis** | | | p = 0.009* |
| Breast cancer | 289 (39.3) | 125 (50.4) | |
| Colorectal cancer | 366 (49.7) | 107 (43.1) | |
| Prostate cancer | 72 (9.8) | 16 (6.5) | |
| **Weight status** | | | p = 0.420 |
| BMI <25 | 292(39.7) | 107(43.1) | |
| BMI ≥25 <30 | 280(38.0) | 87(35.1) | |
| BMI ≥30 | 130(17.7) | 37(14.9) | |
| **Education** | | | p = 0.232 |
| Low-medium | 302 (41.0) | 108 (43.5) | |
| Higher education | 406 (55.2) | 121 (48.8) | |

*significant difference between included and excluded participants.

With the Spearman Brown formula, an ICC >0.80 was achieved with two monitoring days for LPA and three monitoring days for MPA and MVPA (Fig 1). No number of days within the six days timeframe resulted in an ICC >0.80 for VPA due to large intra-individual variance (Table 3 in S1 File).

The ICC and Spearman Brown formula were further calculated stratified on participant characteristics, which revealed some differences in reliability across the subgroups (Fig 2).

An ICC >0.80 was obtained for LPA with three monitoring days in participants with BMI ≥25, low-medium education, colorectal cancer, and age ≥60 years (Fig 2A). For MPA and MVPA, four monitoring days were required in participants with BMI ≥25 and breast cancer (Fig 2B and 2D). The need for additional monitoring days reflected a higher intra-individual variance in PA. For VPA, an ICC >0.80 was found using six monitoring days in females, breast cancer survivors, and participants <60 years, and with four days in participants with BMI ≥30, reflecting lower intra-individual variance in VPA (Fig 2C).

## Discussion

In the present study we assessed the reliability of number of monitoring days representing time in PA intensities in cancer survivors. When accounting for the six-day variation in PA, two monitoring days for LPA, and three monitoring days for MPA and MVPA were sufficient for obtaining reliable estimates. The level of VPA was low, therefore the results for MVPA reflected that of MPA. The low level of VPA and high day-to-day variation within participants suggested that longer monitoring periods are necessary for obtaining reliable estimates. Six monitoring days were close to an ICC >0.80, which could imply that using seven or eight days will exceed the cut-off. However, we did not assess the reliability for a number of days exceeding six days, as our ICC was based on the six days variation. While assessments of more days

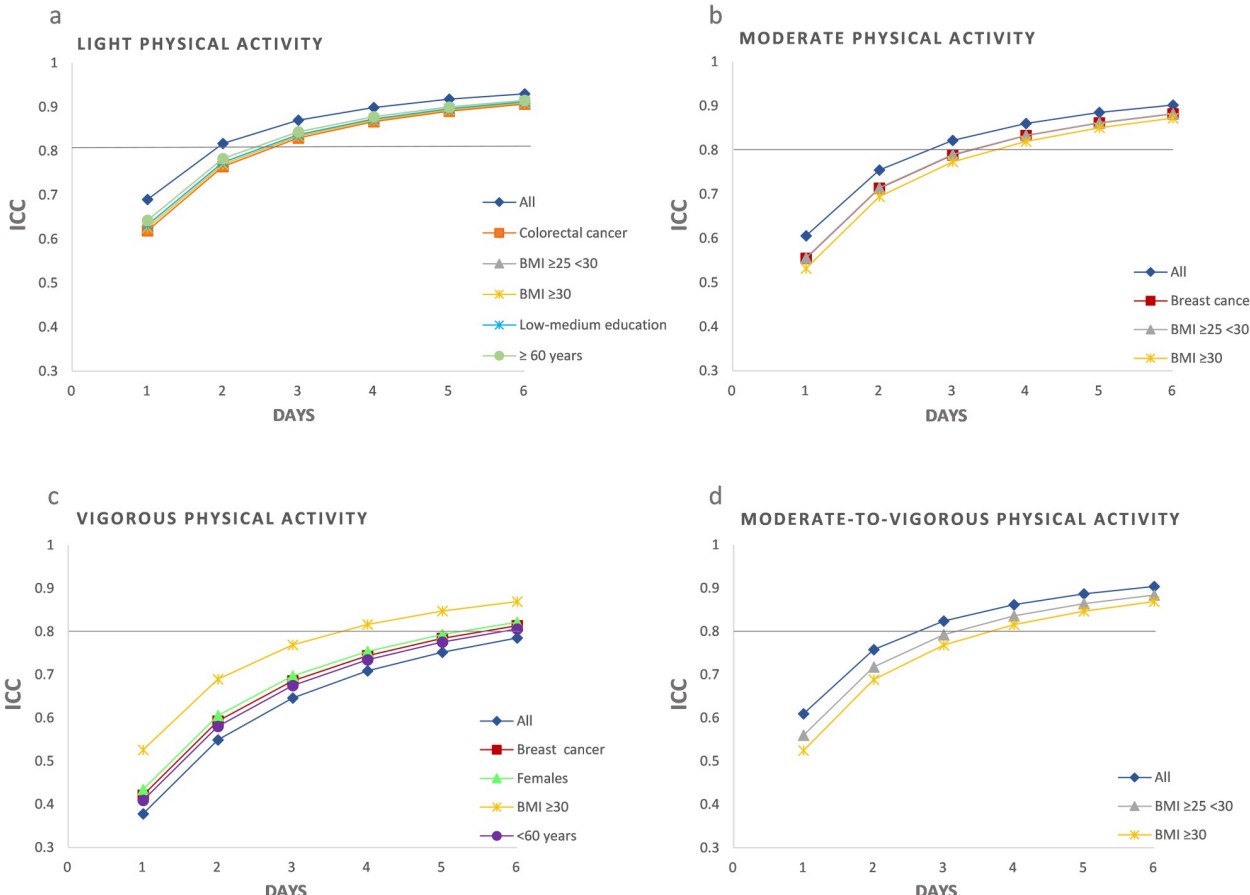

**Fig 2. ICCs for each monitoring day in the total sample and in the stratified samples that deviated from the reliable number of days found for the whole sample of 736 cancer survivors.** (a) Colorectal cancer survivors, participants with BMI ≥25, low-medium education, or ≥60 years old required an additional monitoring day for reliable estimates of LPA. (b) Four monitoring days were required for reliable estimates of MPA in participants with breast cancer or BMI ≥25. (c) While reliable estimates of VPA were not obtained for the total sample, participants with BMI ≥30 achieved reliable estimates with four monitoring days, and females, breast cancer survivors and participants <60 years old achieved reliable estimates with six monitoring days. (d) Participants with BMI ≥25 required four monitoring days for MVPA.

may yield the same variance estimates, thus a similar ICC for one measuring day, it may also increase the intra-individual variance resulting in a lower ICC. In the stratified analyses, some participant characteristics had implications for the variance in PA, thus affecting the reliability.

Our results are in line with findings from a study where SWAM was used in the general population with the same daily wear-time of ≥22 hours [23]. In the study by Scheers et al., three monitoring days were necessary for obtaining an ICC >0.80 for LPA, MPA and MVPA in adults. Similarly, reliable estimates for MVPA using three monitoring days have been found in a smaller sample of colorectal cancer survivors [24]. Compared to the general population, it appears that cancer survivors may have similar or slightly lower intra-individual variability in LPA and MPA but higher variability in VPA [12–22].

To our knowledge, there has been no previous assessments of the impact of sex, age, cancer type, weight status or education on the reliability of monitoring days in cancer survivors. Moreover, underlying explanations for differences in intra-individual variation in PA have not been established. We did not account for external factors such as weather, weekends, or time of year, which could further account for the variance in PA. Such circumstances may have

affected participants with deviating levels of intra-individual variance relative to the total sample, to a lesser or greater extent.

Overweight and obese participants had both higher intra-individual variation in LPA, MPA and MVPA, and significantly lower levels of these intensities compared to normal-weight participants (Table 2 in S1 File). Their level of VPA was also low, but so was their day-to-day variance in VPA. This implies higher proportions of sedentary time and may suggest less planned PA, resulting in sporadic and spontaneous activity throughout the day. On the other hand, engaging in exercise (structured or planned PA) some days of the week can result in higher day-to-day variations compared to individuals who do not exercise. However, the relatively low levels of PA across intensities in overweight and obese participants suggested little engagement in exercise. Higher levels of MPA were associated with being male, having colorectal or prostate cancer, age <60 years, and BMI<25 (Table 2 in S1 File). Higher levels of VPA were associated with age <60 years, higher education, and BMI <25.

The measured activity levels in our study were above the recommended weekly 150 minutes MVPA (Table 2 in S1 File). However, this minimum threshold may not be appropriate when using continuous PA monitoring protocols [14]. In previous studies, researchers have documented how feedback from sophisticated wearable devices worn continuously is incompatible with current PA recommendations and can make people erroneously form the view that they are exceeding recommendations by several fold if adjustments are not made [36]. For MVPA, 1000 minutes weekly, representing 15% of waking time, has been suggested as a more appropriate target when using continuous monitoring [36]. However, it is possible that some participants increased their activity levels as a result of being monitored.

## Strengths and limitations

With this study we were the first to assess the number of monitoring days required for reliable PA estimates in cancer survivors using continuous monitoring. When studying variability in PA levels, having a mixed sample means we may account for more of the variation in PA caused by heterogeneity in the sample. As our sample varied in cancer type, age, sex, socio-economic background, and weight status, together with the large sample size, we may have been able to account for some of the heterogeneity within the cancer population that might cause variations in PA levels. The included variables were chosen based on their availability within the harmonized dataset and their theoretical relevance for PA. However, we did not account for all other relevant covariates which could have further impacted and explained the variation in PA, e.g., treatment type, time since treatment, physical function, fatigue, or cancer stage. Information on treatment type, time since treatment and cancer stage were not sufficient for harmonization. While all participant in the Phys-Can study were assessed before starting neoadjuvant or adjuvant treatment, participants in the CRC-NORDIET study were recruited post curative surgery. About 10% of the CRC-NORDIET participants received pre-surgery radiotherapy or chemoradiotherapy, while about 20% received post-surgery chemotherapy, but lacked information on the duration and number of cycles. This limited the possibility of harmonizing on treatment type and time since treatment. Cancer stage was only available for one study.

Furthermore, we only accounted for the variation in PA across six days, thus reported how well different number of monitoring days represented the observed variation within this timeframe. We do not know whether this variation is consistent across longer monitoring periods. Thus, further research should assess the variability in PA across longer time spans using continuous monitoring in order to establish a reliable number of monitoring days representing longer time periods.

Using cut-offs for acceptable reliability has its limitation and may not be appropriate in all settings. We obtained an ICC of 0.785 for VPA which would have been regarded as sufficient when using a cut-off around 0.7–0.75 as some researchers have previous suggested [16, 34]. All coefficients were listed in (Table 3 in S1 File) under *Inter-individual variance contribution* and can be utilized if different cut-offs are of interest.

## Implications for future research

Researchers should note that some participant characteristics can have implications for the variance in PA affecting how many days some cancer survivors should be monitored in order to obtain reliable estimates. Also, within subject variance in PA can vary independently of PA level. Whether variance in PA and thus the reliability of monitoring days is affected by cancer specific factors including cancer stage, treatment type, symptoms, and late effect, needs further exploring. VPA showed particularly large day-to-day variations within cancer survivors which means that longer monitoring periods may be necessary for obtaining reliable estimates of time spent in VPA. The variation in VPA across longer periods of time and how this affects the reliability should be further assessed. We chose a daily monitor wear-time of ≥22 hours to limit the effect of wear-time on the variance in PA, which has also been applied in previous research using SWAM [23]. However, there is no consensus on how many hours daily a PA monitor should be worn in cancer survivors in order to constitute a valid day, and researchers often use different ways of defining a valid monitoring day [37]. When and how many hours daily PA monitors should be worn to obtain reliable estimates of daily PA, and how the reliability of monitoring days is affected by different wear-time cut-offs should be further explored. PA monitors able to accurately distinguish between non-wear-time, sleep, and sedentary time, should be used to assess the number of days required for reliable estimates of sedentary time in cancer survivors.

## Perspectives

In the present study, 941 (95.63%) cancer survivors had at least three out of seven days with ≥22 hours SWAM wear-time, complying with the minimum of three days found necessary for reliable estimates of LPA, MPA and MVPA. This demonstrates how measurements from more participants relative to the 74.8% complying with the 6-days protocol could have been utilized in a study when assessing their PA levels. Employing a shorter monitoring protocol may possibly facilitate study participation and lower the participant burden. From a researcher perspective, when deciding on an appropriate monitoring period, it should be considered how sex, cancer type, age, education, and weight status are associated with variations in PA. Though intra-individual variance in MPA appears similar to the general adult population, cancer survivors may have lower intra-individual variance in LPA and higher intra-individual variance in VPA.

## Conclusion

In the present study, we assessed the variance in physical activity level across six days with continuous monitoring in breast, colorectal, and prostate cancer survivors 0–9 months post treatment. Based on the observed variance, two monitoring days for light physical activity, and three days for moderate and moderate-to-vigorous physical activity were required for reliable estimates in the total sample. Intra-individual variation in vigorous physical activity was greater and more than six monitoring days appeared necessary for reliable estimates. In the stratified analyses, one additional monitoring day was required for reliable estimates of light physical activity in cancer survivors with colorectal cancer, BMI ≥25, low-medium education,

or age ≥60 years. One additional monitoring day was required for moderate physical activity in cancer survivors with breast cancer or BMI ≥25, while one additional day was required for moderate-to-vigorous physical activity with BMI ≥25. Reliable estimates of vigorous physical activity were obtained for cancer survivors with BMI ≥30, breast cancer, age <60, and for females.

## Supporting information

**S1 File. Contains supporting tables.**
(DOCX)

**S1 Dataset.**
(SAV)

## Acknowledgments

The authors thank the Phys-Can and CRC-NORDIET research teams and all participants for their contribution.

## Author Contributions

**Conceptualization:** Benedikte Western, Ingrid Demmelmaier, Ingvild Vistad, Bjørge Herman Hansen, Andreas Stenling, Sveinung Berntsen.

**Data curation:** Ingrid Demmelmaier, Hege Berg Henriksen, Karin Nordin.

**Formal analysis:** Benedikte Western.

**Methodology:** Benedikte Western, Bjørge Herman Hansen, Andreas Stenling.

**Project administration:** Hege Berg Henriksen, Karin Nordin, Rune Blomhoff, Sveinung Berntsen.

**Supervision:** Ingrid Demmelmaier, Ingvild Vistad, Sveinung Berntsen.

**Visualization:** Benedikte Western.

**Writing – original draft:** Benedikte Western.

**Writing – review & editing:** Benedikte Western, Ingrid Demmelmaier, Ingvild Vistad, Bjørge Herman Hansen, Andreas Stenling, Hege Berg Henriksen, Karin Nordin, Rune Blomhoff, Sveinung Berntsen.

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
