## [Decision Letter · Decision Letter 0]

27 Feb 2023

PONE-D-23-00139How many days of continuous physical activity monitoring reliably represent time in different intensities in cancer survivors?PLOS ONE

Dear Dr. Western,

Thank you for submitting your manuscript to PLOS ONE. After careful consideration, we feel that it has merit but does not fully meet PLOS ONE’s publication criteria as it currently stands. Therefore, we invite you to submit a revised version of the manuscript that addresses the points raised during the review process.

We look forward to receiving your revised manuscript.

Kind regards,

Jianhong Zhou

Staff Editor

PLOS ONE

Journal Requirements:

Additional Editor Comments:

The manuscript was evaluated by two reviewers who provided different comments. Please provide your point-to-point responses to all the comments.

Reviewers' comments:

Reviewer's Responses to Questions

**Comments to the Author**

1. Is the manuscript technically sound, and do the data support the conclusions?

Reviewer #1: Yes

Reviewer #2: No

2. Has the statistical analysis been performed appropriately and rigorously? 

Reviewer #1: Yes

Reviewer #2: No

3. Have the authors made all data underlying the findings in their manuscript fully available?

Reviewer #1: Yes

Reviewer #2: Yes

4. Is the manuscript presented in an intelligible fashion and written in standard English?

Reviewer #1: Yes

Reviewer #2: No

5. Review Comments to the Author

Reviewer #1: This is very interesting study evaluating the number of days with >/= 22 hours of accelerometer wear time needed to evaluate PA in cancer survivors. I have the following questions for the authors that should be addressed in the manuscript:

1. Why wasn't cancer stage, treatment history, and time since completing treatment considered in the analysis? If the data is available, this analysis and findings should be included. If the data is not available and explanation for why it is not available should be included in the manuscript.

2. Conclusions should be revised in order to be specific to the time relevant in the cancer care continuum and cancer type to which this data applies. For example, "Only two days of wear time is needed for LPA in breast, CRC, and prostate cancer survivors at least 2-months post-treatment."

Reviewer #2: There is misunderstanding in the Spearman-Brown formula. K is the ratio of number of new additional days, to the original number of days. So k=6 is actually 6x6=36 days in total. If the authors want to test how many days within 6 days, k should be 1/6, 2/6, …, 6/6 instead.

The analysis needs to be performed again and conclusions needs to be re-written.

To fully understand the variation contributions (intra- or inter), a linear mixed model for longitudinal analysis should be considered. A mean plot by each day and characteristic groups can be illustrated. Furthermore, Bland-Altman plot may be considered too. The authors may consult with a statistician.

ICC better presented with 95% confidence intervals.

In several places, “Analyzes” should be “Analyses”.

6. PLOS authors have the option to publish the peer review history of their article (what does this mean?). If published, this will include your full peer review and any attached files.

Reviewer #1: No

Reviewer #2: No

---

## [Author Response · Author response to Decision Letter 0]

23 Mar 2023

Response to editor comments: 

1. Manuscript has been revised to meet the PLOS ONE style requirements

2. An ethic statement has been added to the Methods section of the manuscript

Respons to reviewer #1:

1. Thank you for the comment. Information about cancer stage, type, and treatment history has been added to the manuscript. 

Unfortunately, we were not able to harmonize data on cancer stage, treatment history and time since diagnosis. Detailed information about cancer stage was only available for one of the included studies. In the Phys-Can study, physical activity was monitored before participants started neoadjuvant or adjuvant treatment, however detailed information about where participants were in their treatment trajectory was not available. In the CRC-NORDIET study, all participants had completed surgery, but some received adjuvant chemotherapy with no additional information on the duration/cycles. Thus, some of these participants may have still been receiving chemotherapy when monitored, while others had finished treatment. This heterogeneity complicated stratification on treatment type and time since treatment. However, some additional information on treatment type has been added to the Method section and Strengths and limitations, page 6 lines 106-107 and page 14 lines 278-285.

2. Conclusion has been revised to be specific of the time relevant in the cancer care continuum and cancer type. 

Response to reviewer #2:

Thank you for the comment, this is certainly something we should specify clearer. If I understand correctly, I believe you are referring to the use of the Spearman-Brown prophecy formula when assessing the number of items in an instrument, and their internal consistency (see de Vet, Mokkink, Mosmuller, and Terwee (2017), section 4). 

The Spearman-Brown formula applied in our study assesses interrater reliability when the number of raters change, see section 4 and appendix A.4 by de Vet et al. (de Vet et al., 2017). In our case a monitoring day would be considered a “rater”. The formula calculates how many scores (days of measurement) need to be average to achieve acceptable reliability. We used the formula for varying days/raters and obtained new coefficients. It is also possible to rewrite the equation and vary desired ICC and obtain values for number of days instead (Mattocks et al., 2008). 

k = [ICCdesired/(1-ICCdesired)][(1-ICCsingle measure)/ICCsingle measure]

However, as we wanted to obtain ICCs for whole days, we applied the current equation.

[k*ICCsingle measure]/[1+(k-1)*ICCsingle measure]

In the formula, the coefficient obtained through the ICC analysis for “single measures” (for example 0.690 for LPA) is plotted into the equation and the number of days (k) varied. As our aim was to assess the absolute agreement between physical activity measures from each day, we used the coefficients representing ICCagreement. 

We have conferred with a statistician (Dr. A. Stenling) and validated our use of the formula against previous research (da Silva et al., 2019; Dillon et al., 2016; Ricardo et al., 2020). If it is of interest, we can send our calculations of the ICC and Spearman-Brown formula. 

We recognize that the ANOVA framework for the ICC has its limitations and that mixed-effect models may be more appropriate in certain settings and conditions, e.g. negative or zero ICC, missing repeated data, confounding effects, sampling error, or large datasets difficult to assess in a limited spreadsheet environment (Chen et al., 2018; Pleil, Wallace, Stiegel, & Funk, 2018). Also, with complex assessments with numerous factors influencing the variance of the assessments, mixed models may be especially relevant. We consider our data to be less subject to such impacts as it is a rather simple analysis of the variance in measured physical activity level with no missing data. 

Previous research comparing spreadsheet calculations, ANOVA and a mixed model approach with both balanced data containing the same number of repeated measures and no missing values, as well as unbalanced data with fewer subjects, resulted in essentially the same ICC estimates. The method of calculation was found to be irrelevant for the simulated data (Pleil et al., 2018). 

While Bland-Altman plots can support visual interpretations of variance between two measures, primarily when comparing with a gold standard method, we do not believe it would contribute to further understanding of our variance estimates. If we were to compare each combination of days against the mean of six days, (as has been previously published (da Silva et al., 2019)) it would require 20 Bland-Altman plots (5 plots for combinations of days times 4 different intensities). Furthermore, while six days served as the criterion, we do not have a defined “gold standard” for number of measuring days, as coefficients for higher number of days may have only small variations in ICC, and not differ from the criterion. 

A modified Bland-Altman plot is possible to create for obtaining one plot within each intensity (4 plots in total), however, the interpretation of these differs slightly from the traditional Bland-Altman plots (Möller, Debrabant, Halekoh, Petersen, & Gerke, 2021). Thus, we regard our choice of method as sufficient for the purpose of the current study.

Confidence intervals have been added to the ICC results.

"Analyzes" has been changed to "analyses".

---

## [Decision Letter · Decision Letter 1]

11 Apr 2023

How many days of continuous physical activity monitoring reliably represent time in different intensities in cancer survivors?

PONE-D-23-00139R1

Dear Dr. Western,

We’re pleased to inform you that your manuscript has been judged scientifically suitable for publication and will be formally accepted for publication once it meets all outstanding technical requirements.

Kind regards,

Justin C. Brown

Section Editor

PLOS ONE

Reviewers' comments:

Reviewer's Responses to Questions

**Comments to the Author**

1. If the authors have adequately addressed your comments raised in a previous round of review and you feel that this manuscript is now acceptable for publication, you may indicate that here to bypass the “Comments to the Author” section, enter your conflict of interest statement in the “Confidential to Editor” section, and submit your "Accept" recommendation.

Reviewer #1: All comments have been addressed

Reviewer #2: All comments have been addressed

2. Is the manuscript technically sound, and do the data support the conclusions?

Reviewer #1: (No Response)

Reviewer #2: (No Response)

3. Has the statistical analysis been performed appropriately and rigorously? 

Reviewer #1: (No Response)

Reviewer #2: (No Response)

4. Have the authors made all data underlying the findings in their manuscript fully available?

Reviewer #1: (No Response)

Reviewer #2: (No Response)

5. Is the manuscript presented in an intelligible fashion and written in standard English?

Reviewer #1: (No Response)

Reviewer #2: (No Response)

6. Review Comments to the Author

Reviewer #1: (No Response)

Reviewer #2: (No Response)

7. PLOS authors have the option to publish the peer review history of their article (what does this mean?). If published, this will include your full peer review and any attached files.

Reviewer #1: No

Reviewer #2: No

---

## [Editor Report · Acceptance letter]

13 Apr 2023

PONE-D-23-00139R1 

How many days of continuous physical activity monitoring reliably represent time in different intensities in cancer survivors 

Dear Dr. Western:

I'm pleased to inform you that your manuscript has been deemed suitable for publication in PLOS ONE. Congratulations! Your manuscript is now with our production department. 

Kind regards, 

on behalf of

Dr. Justin C. Brown 

Section Editor

PLOS ONE